# The Impact of Environmental Regulation, Industrial Structure, and Interaction on the High-Quality Development Efficiency of the Yellow River Basin in China from the Perspective of the Threshold Effect

**DOI:** 10.3390/ijerph192214670

**Published:** 2022-11-08

**Authors:** Xiaoyan Li, Yaxin Tan, Kang Tian

**Affiliations:** 1School of Management and Economics, North China University of Resources and Electric Power, No. 136 Jinshui East Road, Zhengzhou 450046, China; 2College of Information and Management Science, Henan Agricultural University, No. 218, Ping’an Avenue, Zhengzhou 450046, China

**Keywords:** environmental regulation, industrial structure, high-quality development efficiency, SE-SBM model, threshold model

## Abstract

High-quality development efficiency can comprehensively measure the development quality of a region. This study constructed the SE-SBM Model and measured the quality development efficiency of the Yellow River Basin from 2010 to 2019. In panel regression, the periodic effects of industrial structure, environmental regulation, and their interaction terms on the efficiency of high-quality development are analyzed. From the perspective of the threshold effect, we explore the possible threshold of interaction to change the efficiency of high-quality development. The results show: (1) From 2010 to 2019, the high-quality development efficiency of the Yellow River Basin’s watershed segment showed a gradient development trend. (2) In panel regression, the interaction terms positively impact the whole basin’s high-quality development efficiency. However, for different sub-basins, the impact of the core explanatory variable on the efficiency of high-quality development is different. (3) The Yellow River basin has a single significant threshold in the threshold regression. From the perspective of the sub-basin, the upper and lower reaches of the Yellow River have a single significant threshold. There is no threshold in midstream. Based on the research results, the article puts forward relevant suggestions, such as reasonably improving regional imports and exports and introducing high-quality foreign capital, which can provide a basis for relevant departments.

## 1. Introduction

China has a vast territory and several river basins, thus forming many rivers basin economic zones, such as the Yangtze River Economic Belt and the Guangdong-Hong Kong–Macao Greater Bay Area, which are relatively important development strategic areas in China and provide a strong driving force for China’s development [1]. In 2019, Chinese President Xi Jinping emphasized that “protecting the Yellow River is a long-term plan for the great rejuvenation of the Chinese nation” and has hosted several special conferences to study the ecological protection and high-quality development of the Yellow River Basin. As a result, ecological protection and high-quality development in the Yellow River Basin have become a primary national strategy [2].

At present, the regional economy has transformed from a stage of high-speed growth to a stage of high-quality development, with the inherent requirements of quality first and efficiency prioritizing development [3]. Environmental regulation plays an essential role in this transition. Environmental regulation regulates various behaviors that pollute the public environment to protect the ecological environment. The international community has gradually realized the importance of environmental regulation. For example, the United States has controlled pollution from the source [4] and began to gradually transition from implementing mandatory order regulations to incentive policies in the form of rewards and punishments. Japan and Germany arouse the protection awareness of the public and enterprises through incentive mechanisms to achieve longer-term and adequate environmental protection [5,6]. China has incorporated environmental regulation into government performance assessment, including energy conservation, emission reduction, and environmental protection in the performance assessment of local leaders, which can more effectively promote energy conservation, emission reduction, and environmental protection [7].

Appropriate environmental regulation can stimulate innovation, promote technological innovation, and improve efficiency. It can also promote the upgrading of industrial structure through screening effects and spillover effects, thereby affecting the efficiency of high-quality regional development [8]. However, the excessive intensity of environmental regulation may reduce production efficiency and hinder the region’s improvement of high-quality development efficiency. If the intensity of environmental regulation is too low, it is impossible to improve and protect the regional ecological environment, thereby reducing the efficiency of a comprehensive evaluation of high-quality regional development [9]. Appropriate environmental regulation depends not only on the strength but also on the matching of the transmission mechanism to the efficiency of high-quality development [10].

Efficiency is a comprehensive indicator for evaluating high-quality regional development, and social resource allocation needs to be considered. Researchers can analyze the strength of the development level and the space for sustainable development by weighing and comparing regional inputs and outputs. At present, research on high-quality development efficiency generally focuses on the high-quality development efficiency of industries, industries, and resource utilization, such as the high-quality development efficiency of manufacturing, high-quality development efficiency of commercial banks [11], and high-quality efficiency of rural land use [12]. For the high-quality and efficient development of the region, it focuses on a specific aspect of development, such as the high-quality economic development efficiency [13,14] and the high-quality development efficiency of the urban environment [15]. A limited number of pieces of literature discuss the high-quality development efficiency of cities and economic belts [3,16,17].

In terms of efficiency measurement methods, the Stochastic Frontier Analysis (SFA) proposed by Aigner et al. (1977) is often used as a measure of efficiency [18]. Since SFA needs to assume the specific form of the production function, most researchers prefer the data envelopment method (DEA), which introduces the input–output slack problem to avoid bias in the model set [19]. However, the non-radial and non-angular SBM model proposed by Tone (2001) found that this method easily ignores environmental factors and cannot be comprehensively measured by comprehensive indicators [20]. Although these research methods have certain shortcomings, they also lay a research foundation for measuring high-quality development efficiency. Combining the advantages of the super-efficiency DEA model and the SBM model [21,22], this paper adopts the super-efficiency SBM model to measure the high-quality development efficiency of the region comprehensively and comprehensively from the input and output of high-quality regional development combined with undesired output. 

Early studies on environmental regulation, industrial structure, and high-quality development focused on the relationship between environmental regulation, industrial structure, and high-quality development. For example, some studies have found that environmental regulation has a promoting effect on the industrial structure by analyzing the role of environmental regulation in technological progress, international trade, FDI, and other aspects [23,24,25]. Other studies have found that environmental regulation has spatial spillover effects, fixed effects, and spatial transfer effects on industrial structure [26,27,28]. With the rise of high-quality development, scholars have begun to pay more attention to research on the impact of environmental regulation on high-quality development. Most studies believe that environmental regulation can significantly affect the high-quality development of the regional economy through factors such as industrial transformation, transaction costs, technological innovation, human capital, and energy efficiency [29,30,31,32,33,34,35]. In the process of promoting regional economic development, environmental regulation can lead to poverty reduction and enrichment effects, innovation effects, energy effects, adjustment effects, mediation effects, threshold effects, and two-way feedback effects [36,37,38,39,40]. Research on environmental regulation, industrial structure, and high quality currently focuses on the relationship between the three. For example, whether environmental regulation can affect high-quality development through industrial structure [39], environmental regulation and industrial structure have a non-linear relationship with high-quality development [34].

In the current research, there are still the following problems: the efficiency measurement often needs to pay attention to the problem of undesired output; the interaction term between environmental regulation and industrial structure is rarely used as the active path that affects the efficiency of high-quality development. Studies on environmental regulation, industrial structure, and high-quality development efficiency rarely study the non-linear relationship among them. This study adopts the SE-SBM Model that introduces slack variables, which can be solved in a larger feasible region. We consider the undesired output in the development of the river basin while also addressing the efficiency ordering problem between effective decision-making units and comprehensively and dynamically measuring the high-quality development efficiency of the Yellow River Basin. Second, we constructed a panel data model to analyze the impact of environmental regulation, industrial structure, and their interaction on the efficiency of high-quality development. We theoretically analyze the non-linear relationship between environmental regulation, industrial structure, and their interaction terms on high-quality development efficiency and determine the threshold value from the whole watershed and sub-watershed. Finally, according to the research results, we put forward countermeasures and suggestions to promote the high-quality development of the river basin.

## 2. Materials and Methods

### 2.1. Overview of the Study Area

The Yellow River Basin covers a vast area. It flows through nine provinces in China, namely Qinghai, Sichuan, Gansu, Ningxia (upper area), Inner Mongolia, Shanxi, Shaanxi (midstream), Henan, and Shandong (downstream). The Yellow River Basin is a prominent grain-producing area with significant energy, chemical, raw material, and industrial bases in China. It is also one of the essential protection areas for the country’s energy and environment. The basin pays attention to environmental regulation. It has issued policy documents such as the “Plan for Ecological Environmental Protection in the Yellow River Basin” and “Outline of the Plan for Ecological Protection and High-Quality Development in the Yellow River Basin.” The provinces in the basin have also introduced environmental regulations suitable for their development.

The Yellow River Basin attaches great importance to optimizing and adjusting industrial structures. However, from the industrial development data of 9 provinces in the Yellow River Basin (2019 data): the proportion of the primary industry is higher than the national average [41,42], the contribution rate of the primary industry to the regional GDP has always been low. Except for Sichuan Province and Gansu Province, the contribution rate of the primary industry to the regional GDP in other provinces (autonomous regions) is less than 10%. The Yellow River Basin is a crucial energy production base in China, with abundant natural and mineral resources, a robust industrial base, and obvious development advantages. The GDP of the secondary industry in the basin accounts for 41% of China’s GDP, and it is also a significant driving force for economic growth in the basin. The development of the tertiary industry in the Yellow River Basin is slow. The tertiary industry in the Yellow River Basin accounted for 44.4%, lower than the national average of 53.3% and the Yangtze River Basin of 54.1%. The tertiary industry in the Yellow River Basin only accounts for 23.4% of the national tertiary industry, which is on a downward trend. The industrial development in the Yellow River Basin has yet to form its distinct advantages, and the development level of the tertiary industry needs to be further improved [43]. The data in this paper come from the 2010–2019 provincial statistical yearbooks, national databases, and EPS databases.

### 2.2. SE-SBM Model

Efficiency is analyzed from the perspectives of input and output and considers the allocation of social resources to evaluate the comprehensive level of high-quality regional development. Efficiency measurement can effectively evaluate the high-quality development efficiency of the region, which is conducive to the horizontal comparison between regions and finding the natural source of power or the crux of the problem.

In the SE-SBM model, it is assumed that there are m decision-making unit (DMU), each DMU has m inputs denoted as $X_{i}$, and each input has a weight of $V_{i}$, where i can be taken from 1 to m. Each decision-making unit has q outputs, denoted as y, and each output has a weight of $U_{r}$, where i can be taken from 1 to q. According to the efficiency value solved by the Model, the result can evaluate the efficiency level of each decision-making unit and its importance in the Model.

Regarding selecting measurement indicators for high-quality development, refer to Zhang (2021)’s [16,44,45,46] research on high-quality development. The indicators for selecting input and output are shown in Table 1.

### 2.3. Panel Regression Model

Panel regression models are widely used to analyze whether there is a significant relationship between independent and dependent variables. They can also analyze the strength of the influence of multiple independent variables on a dependent variable [47,48]. This study uses panel regression to explore the correlation between each variable and the high-quality development efficiency of the Yellow River Basin, laying the foundation for constructing the threshold model in the following. The core variables of this paper are environmental regulation, industrial structure, and the interaction of environmental regulation and industrial structure. Unlike previous studies, this study introduces the interaction between environmental regulation and industrial structure. It explores whether environmental regulation can affect the efficiency of high-quality regional development by affecting the industrial structure. If the influence result is positive, it means that the interaction between environmental regulation and industrial structure can promote the high-quality development efficiency of the Yellow River Basin. Otherwise, it means that the interaction between environmental regulation and industrial structure inhibits the high-quality development efficiency of the Yellow River Basin. The panel model is set as follows:(1)$$E_{it}=\beta_{0}+\beta_{1}ER_{it}+\beta_{2}PC_{it}+\beta_{3}ER_{it}*PC_{it}+\beta_{4}X_{it}$$
where i, t represent region and time respectively, $\beta_{0}$ represents the individual characteristics of the observed value, $E_{it}$ represents the high-quality development efficiency value of i province in t year, $ER_{it}$ represents the environmental regulation intensity of i province in t year. The intensity of environmental regulation is expressed as the proportion of investment in industrial pollution control to regional GDP, $PC_{it}$ represents the industrial structure level of i province in t year, $ER_{it}*PC_{it}$ is the interaction term between environmental regulation and industrial structure of i province in t year. X represents the control variable introduced. ε is the random disturbance term. 

Many factors affect the efficiency of high-quality development in a region. Referring to existing research, we selected the indicators of opening-up level, population density, and economic development level as control variables: index of the level of opening to the outside world (IAE): improving the level of opening to the outside world can promote the adequate flow of resources; population density index (PD): the regional population density is relatively high, indicating that the population can be effectively attracted, indicating that the regional development situation is good; economic development level (GDP): the improvement of economic development level will promote the increase in the high-quality development efficiency value.

### 2.4. Threshold Model

Panel regression analysis verifies the impact of the interaction between environmental regulation and industrial structure on the efficiency of high-quality development. However, considering that industrial development is profitable, environmental regulation and regional high-quality development efficiency are non-profit, there is a theoretical nonlinearity between the two, also known as the theoretical threshold effect. The threshold model can measure the non-linear phase relationship between explanatory variables and explained variables. According to Hansen’s threshold estimation method, the primary model of the threshold effect is set as follows:(2)$$Y_{it}=\mu_{i}+\beta_{1}X_{it}I(q_{it}\geq ϒ)+\beta_{2}X_{it}I(q_{it}\geq ϒ)+\varepsilon_{it}$$
where i and t represent region and time respectively; $Y_{it}$ Represents the explained variable; I represents the indicative function, ϒ represents the threshold value, $q_{it}$ represents the threshold variable; ε denotes the random disturbance term. According to Hansen’s basic threshold model and combined with the research in this paper, the Model is constructed as follows:(3)$$E_{it}=\beta_{0}+\beta_{1}ER_{it}+\beta_{2}PC_{it}+\beta_{3}ER_{it}*PC_{it}I(q_{it}\geq ϒ)+\beta_{4}ER_{it}*PC_{it}I(q_{it}<ϒ)+\beta_{5}X_{it}+\varepsilon_{it}$$

The explained variable $E_{it}$ represents the high-quality development efficiency value of i province in t year. Core explanatory variables: industrial structure $PC_{it}$, environmental regulation $ER_{it}$, the interaction term between environmental regulation and industrial structure $ER_{it}*PC_{it}$, and the selection of control variables are the same as formula (1). Using Stata software, the threshold effect test was carried out through the panel model introducing time, region, and other observation variables. 

## 3. Results and Discussion

### 3.1. Results and Discussion on High-Quality Development Efficiency of the Yellow River Basin

According to the relevant data of nine provinces in the Yellow River Basin from 2010 to 2019, we processed the incomplete year value in the indicator and set the weight ratio of expected output to undesired output as 1:1 [49]. MaxDEAPro software was used to calculate the efficiency value of high-quality development in the Yellow River Basin from 2011 to 2019. The results are shown in Table 2.

China divides the economy of the Yellow River Basin into upstream, midstream, and downstream divisions. The calculation results show that the regional high-quality development efficiency is related to the sub-watershed segment. According to the calculation results of the efficiency of each province in the Yellow River Basin (take the average of the calculated results for each province), the high-quality development efficiency value of the sub-basin of the Yellow River can be obtained, as shown in Table 3.

From the results in Table 1, the high-quality development efficiency levels of Shaanxi Province, Gansu Province, Qinghai Province, Sichuan Province, and Inner Mongolia Autonomous Region are relatively low. The high-quality development efficiency levels of Henan Province and Shandong Province are in the leading position in the Yellow River Basin. The main reason is that Shaanxi Province, Gansu Province, and other provinces are in the middle and upper reaches of the Yellow River Basin, and location conditions restrict their economic development and technological innovation. Henan Province and Shandong Province are close to the developed coastal areas in the east and can use their advanced technological advantages to promote the optimization, transformation, and upgrading of the industrial structure and bring convenient resources and driving forces for their development [50]. The conclusions of Xu et al. are contrary to ours. This is because the upstream region is an important ecological barrier area in China. The state is financially supported and pays attention to protecting the ecological environment so that high-quality development is better. The economic foundation of the downstream region has developed rapidly, attracting many foreign populations, and the agglomeration of the population exceeds a specific carrying capacity, resulting in poor regional high-quality development [51,52].

Figure 1 shows that the development efficiency values of the upper, middle, and lower reaches generally fluctuate. Among them, from 2015 to 2017, the high-quality development efficiency of various watersheds showed a relatively apparent decline. The reason may be that the global economy was affected by political uncertainty and the escalation of trade and currency wars during this period. In addition, all aspects of social development have been significantly impacted, and the economic development of the entire Yellow River Basin has been affected. However, with new goals and strategies for China’s economic development and the gradual improvement of the international situation, the Yellow River Basin has gradually recovered and developed, showing a relatively stable trend. By comparing previous studies, we found that there is a deviation from the conclusions of Zhou et al., which may be because the high-quality development of the watershed pays more attention to the dimensions of power conversion, structural optimization, achievement sharing, and environmental protection [50]. In addition, the high-quality development in the middle and lower reaches is more volatile, and the upstream region is less volatile. The main reasons are the developed downstream economy, rapid technological progress, and rapid development of high-quality development efficiency. The middle reaches may be due to the introduction of the high-quality development plan for the Yellow River Basin, which has increased financial policy support, and the high-quality development efficiency has developed better. In the upstream region, due to resource endowment, underdeveloped economic development, relatively backward technology, and small fluctuations in high-quality development efficiency [52].

### 3.2. Results and Analysis of Panel Model Regression

In the panel model, the LLC and Fisher-ADF tests were first used to test the stationarity of the panel data to avoid spurious regression. The test results showed that all data’s LLC and Fisher-ADF tests were consistent and passed the stationarity test. Panel regression analysis was performed using software Eviews to obtain the impact results of each variable on the high-quality development efficiency of the entire Yellow River Basin, upstream, midstream, and downstream basins. The specific results are shown in Table 4.

#### 3.2.1. Effects of Environmental Regulation, Industrial Structure, and Interaction Terms on the High-Quality Development Efficiency in the Whole Watershed

From the perspective of the whole basin, the impact of environmental regulation on the high-quality development efficiency of the Yellow River Basin (1.70 × 10^−7^) has a weak role in promotion. In contrast, the impact of industrial structure on high-quality development efficiency is relatively high (0.239), indicating that the upgrading and optimization of industrial structure has a noticeable effect on the development of the whole basin. The interaction term of environmental regulation and industrial structure has a positive effect on the high-quality development efficiency of the Yellow River Basin. It shows that environmental regulation can promote the high-quality development efficiency of the Yellow River Basin through the industrial structure. 

Environmental regulation has little impact on the efficiency of high-quality development in the Yellow River Basin. In contrast, industrial structure upgrading has a more significant impact on the efficiency of high-quality development. The possible reason is that the high-quality development of the Yellow River Basin is relatively backward in China. Environmental regulation follows the cost theory and is limited in promoting high-quality development efficiency. The upgrading of the industrial structure can promote the coordinated development of the three industries and extend the industrial chain, thereby promoting the high-quality development efficiency of the Yellow River Basin. 

The interaction between environmental regulation and industrial structure positively impacts the efficiency of high-quality development. Environmental regulation can effectively improve the technological innovation capability of enterprises, promote the upgrading of regional industrial structure, and thus improve the efficiency of high-quality development in the region. However, due to the difference between rationalization and advanced industrial structure, the current industrial structure of the Yellow River Basin only tends to be rationalized, and its impact on the efficiency of high-quality development is limited [53]. It also causes the interaction between environmental regulation and industrial structure to have a weak positive impact on the high-quality development efficiency of the whole basin [39].

#### 3.2.2. Effects of Environmental Regulation, Industrial Structure, and Interaction Terms on High-Quality Development Efficiency in Sub-Watershed

From the perspective of the sub-watershed, environmental regulation has a weak positive impact on the midstream region’s high-quality development efficiency (0.004). In contrast, industrial structure has a significant positive impact on high-quality development efficiency (0.913). The interaction between industrial structure and environmental regulation has a more significant positive impact on high-quality development efficiency (2.163). Environmental regulation can significantly affect regional high-quality development efficiency through the industrial structure. Environmental regulation has a significant positive impact on the high-quality development efficiency (6.047) in the midstream region, and industrial structure also has a positive impact on the high-quality development efficiency (1.271) in the midstream region. However, industrial structure and environmental regulation interaction negatively impact high-quality development efficiency (−5.746). Environmental regulation inhibits high-quality development efficiency through the industrial structure in the midstream region. Environmental regulation positively impacts downstream regions’ high-quality development efficiency (1.721). Industrial structure has a relatively small impact on the high-quality development efficiency (0.547), and the interaction term between environmental regulation and industrial structure significantly impacts the high-quality development efficiency (1.801). Environmental regulation can effectively promote the improvement of high-quality development efficiency through the industrial structure of the midstream region.

In the above analysis, environmental regulation significantly impacts the middle and lower reaches of the Yellow River but less on the upper reaches. Possible reasons are that China’s management resources for the middle and lower reaches of the Yellow River are highly concentrated, which makes environmental regulation high-quality in the middle and lower reaches of the Yellow River. The impact of development efficiency is significant. The upstream is in the west, the economic development is relatively weak, and some areas are essential energy bases, which puts the upstream environment in a state of high load and severe environmental pollution. In addition, environmental regulation increases the cost of regional governance, fails to release the policy dividends of environmental regulation of the upstream region, and does not reflect the promotion of its high development efficiency [54]. 

In the midstream region, the interaction between environmental regulation and industrial structure negatively impacts the efficiency of high-quality development. The main reason is that the environment in the middle reaches of the Yellow River is relatively harsh, the contradiction between environmental optimization and industrial development is relatively sharp, and environmental regulation and industrial structure present a situation of mutual restriction. Therefore, the environmental governance of the midstream is still an urgent task [55]. For upstream and downstream, the environment and economic development are relatively coordinated, so the interaction terms of environmental regulation and industrial structure positively impact their watersheds.

#### 3.2.3. The Influence of Other Control Variables on the High-Quality Development Efficiency

The opening level of the whole Basin has the highest impact on the high-quality development efficiency of the Yellow River Basin (7.886), indicating that the high-quality development efficiency is greatly affected by the opening level in the whole Yellow River Basin. In the upstream and downstream, the impact of opening to the outside world on the high-quality development efficiency is 1.900 and 0.724, respectively, indicating that improving the opening level of the upstream and downstream regions can promote the improvement of high-quality development efficiency to a certain extent. However, in the midstream region, the influence of opening to the outside world is negative (−0.474), indicating that the improvement of opening to the outside world will inhibit high-quality development efficiency. Therefore, the midstream region’s opening level to the outside world needs to be reasonably controlled in the development process. The impact of opening level to the outside world on the high-quality development efficiency of the entire Yellow River Basin is more evident than that of the sub-basins. Due to the scale effect of opening to the outside world, when the level of opening to the outside world is generally improved in the entire watershed, the industrial scale of the region will expand accordingly, and external competition will promote efficiency [56].

For the sub-basins, the opening to the outside world has a limited effect on the efficiency of high-quality development, especially in the middle reaches showing a negative effect. The possible reason for this situation is that the economic development level of the middle reaches is average. Suppose the degree of opening to the outside world is increased. In that case, pollution may be transferred, or it may impact the industry in the region, which will reduce the efficiency of high-quality development in the basin. The reality is that most enterprises in the lower reaches of the Yellow River are technology-intensive, and most enterprises in the upper and middle reaches of the Yellow River are labor intensive. Opening to the outside world has different promotion effects on labor-intensive and technology-intensive enterprises, which leads to different promotion effects on high-quality regional development [57].

### 3.3. Results and Analysis of Threshold Model Regression

Stata15.0 software is used to analyze whether there is an obvious inflection point between the interaction term of industrial structure and environmental regulation on the high quality-development efficiency. Due to the limited sample of this paper, the three-threshold model is not considered temporarily. Data analysis results show that the threshold variable has a single threshold effect and passes the 5% significance test, and the threshold value of the industrial structure is 47.63. Therefore, a single threshold model was constructed to further explore the relationship between threshold variables and high-quality development efficiency of the Yellow River Basin, and regression and analysis results were obtained, as shown in Table 5.

When the industrial structure is at different levels, the impact of the interaction term between environmental regulation and industrial structure on the high-quality development efficiency of the Yellow River Basin passes the significance test of 1%, indicating that the whole basin has a threshold effect. When the industrial structure is lower than 47.63, the impact of the interaction term between environmental regulation and industrial structure on high-quality development efficiency is negative (−3.313). When the industrial structure has not reached the threshold value, environmental regulation has a tremendous negative impact on the high-quality development efficiency through the optimization and upgrading of the industrial structure. When the industrial structure is on the right side of the threshold value, the regression coefficient is slightly smaller but still negative (−2.860). Its negative influence is slightly weakened, but the change range is small. The industrial structure of the Yellow River’s upper, middle, and lower reaches is relatively simple. For example, the upstream area is dominated by ecological industries, the energy industry dominates the midstream area, and the downstream area is dominated by manufacturing. Therefore, it is difficult for upgrading a single industrial structure to affect the development efficiency of the whole basin positively.

When the industrial structure index is on the right side of the threshold value, the interaction term of industrial structure and environmental regulation has a less negative impact on the efficiency of high-quality development [58]. The possible reason is that through the adjustment of the industrial structure in the Yellow River Basin, the proportion of tertiary industry has increased, and the pollution has been relatively reduced. Suppose the development of the industrial structure exceeds the threshold value. In that case, the industrial structure is gradually advanced, and the learning of foreign advanced technology is emphasized, which can increase productivity and reduce the interaction items’ negative impact on the efficiency of high-quality development [39]. 

Overall, environmental regulation can play a significant role in high-quality development efficiency through industrial structure, but its threshold change needs to be apparent. Considering that the high-quality development efficiency value of the Yellow River Basin has obvious gaps due to different basin segments, we have carried out an analysis of the threshold effect of the Yellow River Basin by sub-basin, as shown in Table 6. The threshold test results of the sub-watershed are obtained: The *p*-value of the upstream basin is within the 95% confidence interval of the threshold value, and the significance test of 5% is passed. The *p*-value of the downstream watershed is within the 95% confidence interval of the threshold value, and the significance test of 1% is passed. When the industrial structure is used as the threshold variable, the interaction term of industrial structure and environmental regulation significantly affects the high-quality development efficiency of the upstream and downstream regions. However, the *p* value did not pass the significance test in the midstream, so there was no threshold effect. The significance level of the interaction term between industrial structure and environmental regulation on high-quality development efficiency was analyzed upstream, midstream, and downstream under the condition that industrial structure was used as the threshold variable.

As for the upstream area, when the industrial structure is on the left of the threshold value (44.50), the *p*-value passes the significance test of 10%, and the impact of the interaction term between industrial structure and environmental regulation on high-quality development efficiency is 1.740. However, the *p*-value fails the significance test when the industrial structure is higher than the threshold (44.50). The interaction term has no significant effect on the high-quality development efficiency. As for the midstream, no matter whether the *p*-value is on the left or right side of the threshold value, the significance test did not pass, so there was no significant threshold effect. As for the downstream, when the industrial structure is lower than the threshold value (47.63), the *p*-value passes the significance test of 5%. The influence coefficient of the interaction term between industrial structure and environmental regulation on the high-quality development efficiency is −2.98. When the industrial structure is higher than the threshold value, *p* values pass the significance test of 5%, and the influence coefficient of the interaction term on high-quality development efficiency is −2.57. The threshold effect of the downstream watershed is very significant. Regardless of whether the industrial structure is on the left or right side of the threshold value, the interaction term of industrial structure and environmental regulation negatively impacts the efficiency of high-quality development.

It can be seen from the analysis that when the upstream industry structure is on the left side of the threshold value (44.50), the interaction term between industry structure and environmental regulation has a significant threshold effect on high-quality development efficiency. The reason may be that the scale and speed of the upstream economy are relatively backward. Environmental regulation promotes the upgrading and optimization of the industrial structure. With the rationalization and advancement of the industrial structure, high-quality economic and human resources are introduced, and the high-quality development efficiency of the upstream area is significantly promoted [59]. When the industrial structure is on the right side of the threshold value (44.50), the interaction term of industrial structure and environmental regulation has no significant effect on the high-quality development efficiency. The reason may be that the upstream area is restricted by location conditions, technical level, and economic foundation. After the industrial structure develops to a certain level, it will face the bottleneck of advanced industrial development, making it unable to significantly impact the upstream high-quality development efficiency [16]. 

As for the midstream area, there is no significant threshold effect. The midstream area is deep inland; its development mainly depends on agriculture, energy, and raw material industries, and their added value is low. Therefore, single industrial structure optimization and upgrading cannot have a positive transmission effect on the midstream. High-quality development efficiency cannot be boosted by the interaction term of industrial structure and environmental regulation [14]. 

As for the downstream area, when the industrial structure is on the left of the threshold value (47.63), the impact of the interaction term between industrial structure and environmental regulation on the high-quality development efficiency is −2.98. When the industrial structure is on the right side of the threshold value (47.63), the impact of the interaction term between industrial structure and environmental regulation on the high-quality development efficiency is −2.57. The interaction term negatively impacts high-quality development efficiency, whether the industrial structure is on the left or right side of the threshold value. The reason may be that the downstream region is near the eastern coastal developed areas. High-quality human and economic resources can be absorbed; however, due to the aggregation effect of high-quality resources, downstream energy consumption pollution is increased, and the level of high-quality development efficiency is hindered [19]. When the industrial structure is on the right of the threshold, the interaction term negatively impacts high-quality development efficiency. However, the lower starting point of economic development is higher, resulting in a more negligible impact on the right side of the threshold than on the left side. After the economic level develops to a certain level, upgrading the industrial structure depends on technological progress. Technological progress will promote the improvement of production efficiency, improve the status of environmental pollution, and weaken the negative impact of the interaction term on the high-quality development of downstream areas.

## 4. Conclusions

This study focused on the impact of environmental regulation, industrial structure, and interaction on the high-quality development efficiency of the Yellow River Basin. We obtained the following conclusions and provide corresponding recommendations based on the conclusions. 

Although the development level of the Yellow River Basin fluctuates, it is generally on a rising trend. The development process of different watershed segments presents stages and heterogeneity. From the perspective of sub-basins, the high-quality development efficiency of the downstream areas of the Yellow River Basin is significantly higher than that of the middle and upper reaches. The level of economic development and financial investment plays an essential role in the efficiency of high-quality development. The research suggests that we should improve economic development and increase financial investment.

From the analysis results of panel regression results, we can see that the interaction between environmental regulation and industrial structure dramatically impacts the efficiency of high-quality development, and the interaction between upstream and downstream environmental regulation and industrial structure has a more significant impact. From the regression results of the control variables, there is heterogeneity in the influence of each control variable on the high-quality development efficiency of the Yellow River Basin. The level of opening to the outside world has a positive effect on the efficiency of high-quality development. In contrast, population density and regional GDP negatively affect the efficiency of high-quality development. The study believes that in the process of high-quality development of the Yellow River Basin, regional import and export levels should be reasonably improved. Second, the introduction of high-quality foreign capital to promote regional economic development should take place. Finally, enterprises in the basin should be encouraged to develop into resource-saving and environment-friendly enterprises.

The analysis of the threshold regression results shows that the interaction term of the industrial structure and environmental regulation of the whole basin has a significant single threshold effect on the high-quality development efficiency of the Yellow River Basin. The analysis of the threshold regression results of the sub-basins shows that when the upstream industrial structure is on the left side of the threshold value, the interaction term of industrial structure and environmental regulation has a significant role in promoting high-quality development efficiency. At the same time, there is no threshold effect on the right side of the threshold value. The threshold effect of high-quality development in downstream areas is very significant. Whether the industrial structure is on the left or right side of the threshold, the interaction between the industrial structure and environmental regulation negatively impacts the efficiency of high-quality development. In this regard, we put forward the following suggestions: the upstream area needs to improve the technical level, optimize the industrial structure, and promote the rationalization and advanced nation of the industrial structure. In the development process of the midstream region, it is necessary to effectively control the population density and reduce the adverse environmental effects caused by population agglomeration. Since there is no significant threshold effect in the midstream region, it is only necessary to focus on the diversified development of industries and change the structure of a single industry.

## Figures and Tables

**Figure 1 ijerph-19-14670-f001:**
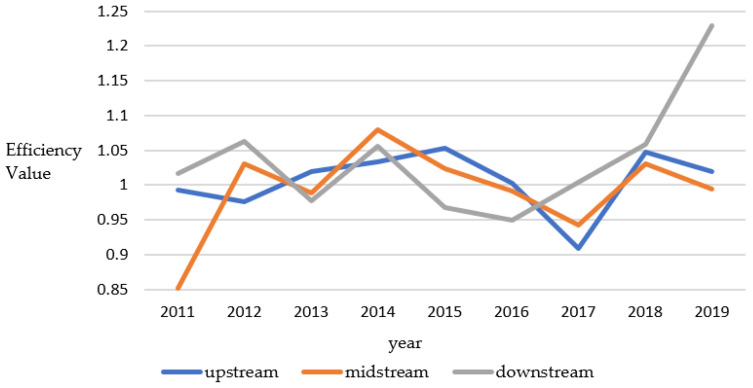
Line graph of the results of high-quality development efficiency measurement in the Yellow River sub-basin.

**Table 1 ijerph-19-14670-t001:** The Yellow River Basin Province/City high-quality development efficiency index system.

Index Type	Index Classification	Basic Index	Unit	Index Attribute
Input index	Labor input	Employed persons in urban units	Ten thousand yuan	+
	Fiscal expenditure	General budget expenditure	Ten thousand yuan	+
	Capital input	Total investment of foreign-invested enterprises	Millions of U.S. dollars	+
	Energy input	Total water resources	Billion cubic meters	+
Desirable output	Industrial structure	The proportion of the tertiary industry in the gross regional product	The percentage	+
	Economic development	Per capita gross regional product	Yuan/person	+
	Medical and healthcare	Number of health facilities	Unit	+
	Technological progress	Technical market turnover	Ten thousand yuan	+
Undesirable output	Unemployment rate	Registered urban unemployment rate	The percentage	-
	Environmental pollution	Total wastewater discharge	Ten thousand tons	-

**Table 2 ijerph-19-14670-t002:** Calculation results of high-quality development efficiency of provinces in the Yellow River Basin.

Province/Year	2011	2012	2013	2014	2015	2016	2017	2018	2019
Shanxi	0.808	1.043	0.906	1.041	1.052	1.005	1.009	1.019	1.063
Inner Mongolia	0.887	0.913	1.032	1.198	1.003	1.069	1.105	1.014	1.008
Shan Dong	0.874	1.067	0.894	1.246	1.006	1.004	1.005	1.014	1.203
Henan	1.158	1.059	1.061	0.865	0.929	0.896	1.003	1.103	1.254
Sichuan	1.037	0.872	1.043	1.012	1.042	1.003	0.925	1.003	1.019
Shaanxi	0.858	1.137	1.026	1.002	1.016	0.902	0.716	1.061	0.910
Gansu	0.917	0.909	1.018	1.089	1.054	1.005	1.002	1.013	1.006
Qinghai	1.003	1.028	1.097	1.004	1.089	0.983	0.938	1.002	1.013
Ningxia	1.014	1.105	0.923	1.032	1.025	1.016	0.770	1.173	1.042

**Table 3 ijerph-19-14670-t003:** Calculation results of sub-basin efficiency for high-quality development in the Yellow River Basin.

Watershed Segment	2011	2012	2013	2014	2015	2016	2017	2018	2019
upstream	0.993	0.976	1.020	1.034	1.053	1.002	0.909	1.048	1.020
midstream	0.851	1.031	0.988	1.080	1.024	0.992	0.943	1.031	0.994
downstream	1.016	1.063	0.978	1.056	0.968	0.950	1.004	1.059	1.229

**Table 4 ijerph-19-14670-t004:** The regression results of variables on the high-quality development efficiency of the Yellow River Basin ^1^.

Variable Name	The Whole Basin	Upstream	Midstream	Downstream
ER	1.70 × 10^−7^ (1.98) **	0.004 (20.91) ***	6.047 (9.69) ***	1.721 (6.35) ***
PC	0.239 (1.99) **	0.913 (5.22) ***	1.271 (8.99) ***	0.547 (2.60) **
ER∗PC	0.163 (1.37)	2.163 (3.39) ***	−5.746 (−8.29) ***	1.801 (6.27) ***
IAE	7.886 (3.31) ***	1.900 (4.01) ***	−0.474 (2.91)	0.724 (3.38) ***
UPD	−0.350 (−0.98)	0.745 (5.39) ***	1.135 (2.91) ***	0.536 (1.69) ***
GDP	−2.79 × 10^−5^ (−2.71) ***	2.245 (2.86) ***	1.179 (3.07) ***	0.834 (4.01) ***

^1^ The value in parentheses is the critical value; values outside parentheses are regression coefficients; ***, ** respectively represent the significance level of 1%, 5%.

**Table 5 ijerph-19-14670-t005:** Basin-wide panel threshold regression results.

Explanatory Variable	Coefficient	Critical Value	Prob
ER	3.218	3.760	0.000
PC	0.008	1.780	0.080
GDP	−1.489	−2.860	0.006
UPD	0.144	1.060	0.294
IAE	0.090	0.980	0.332
ER × PC (PC < 47.63)	−3.313	−3.590	0.001
ER × PC (PC > 47.63)	−2.860	−3.460	0.001

**Table 6 ijerph-19-14670-t006:** Regression results of threshold core variables in sub-basin threshold panel.

Sub-Watershed	Explanatory Variable	Coefficient	Threshold	Prob
upstream	ER × PC (PC < 44.50)	1.740	3.784	0.090
	ER × PC (PC > 44.50)	1.450	3.310	0.160
midstream	ER × PC (PC < 49.70)	−0.890	2.957	0.384
	ER × PC (PC > 49.70)	−0.780	2.740	0.444
downstream	ER × PC (PC < 47.63)	−2.980	1.621	0.016
	ER × PC (PC > 47.63)	−2.570	1.622	0.030

## Data Availability

The data that support the findings of this study are available from the corresponding author upon request.

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
