# Peer review of "The Impact of Environmental Regulation, Industrial Structure, and Interaction on the High-Quality Development Efficiency of the Yellow River Basin in China from the Perspective of the Threshold Effect"

_ijerph, 2022, doi:10.3390/ijerph192214670_

Round 1

Reviewer 1 Report

This paper shows a structured analysis of the high-quality development efficiency of the Yellow River Basin. The writing is clear and the adopted method is solid. However, there are some issues that need to be addressed before publishing.

1. Some theoretical conclusions sound arbitrary to me. For instance, pp 3 states that  "The development of the tertiary industry in the Yellow River Basin is slow", but why? Some references are needed.

2, Why this paper adopted the particular method SE-SBM Model? There are other methods there, what is your justification for your methodology? 

3, Measurement indicators only adopt Zhang's(2021) work, why?

4, The same holds true for the pannel regression model, why choose it?

5, There are format issues that need to be proofread, for instance, pp2 "A limited number of pieces of literature discuss the high-quality development efficiency of cities and economic belts [Error! Bookmark not defined.,8]."

Author Response

Dear Reviewer,

Thank you for your comments concerning our manuscript entitled “The impact of environmental regulation, industrial structure, and interaction on the high-quality development efficiency of the Yellow River Basin in China from the perspective of the threshold effect." Those comments are all valuable and helpful for revising and improving our paper. We have studied the comments carefully and have made corrections that we hope meet with your approval. Revised portions are marked in yellow on the paper. The primary corrections to the paper and point-to-point responses to your comments are as follows: 

Responses to the reviewer’s comments:

Comment 1: Some theoretical conclusions sound arbitrary to me. For instance, pp 3 states that "The development of the tertiary industry in the Yellow River Basin is slow", but why? Some references are needed.

Response: Thanks a lot for your comment. We mainly analyze the reasons for the slow development of the tertiary industry in the Yellow River Basin from the following aspects:

  1. Analysis of the proportion of the tertiary industry in the Yellow River Basin. The latest 2021 data shows that the proportions of the three industrial structures in the Yellow River Basin are 10.9:41.4:47.7, respectively. The data from 2011 to 2019 are selected in the article. The past years' data show that the tertiary industry is at most 50% (the proportion of the tertiary industry in the Yellow River Basin in 2019 was only 44.4%).
  2. Comparative analysis with the national average level and the Yangtze River Basin. The proportion of the three industrial structures in the whole country and the Yangtze River Basin is 7.3:39.4:53.3 and 7.6:38.3:54.1, respectively. The proportion of the tertiary industry in the whole country and the Yangtze River Basin is significantly higher than that in the Yellow River Basin, and the proportion of the tertiary industry in the Yellow River Basin is only 23.4% of the national tertiary industry. To sum up, it is concluded that the development of the tertiary industry is relatively slow.

In response to your comment, the authors supplement relevant data in the manuscript to illustrate that the development of the tertiary industry in the Yellow River Basin is relatively slow. The added contents are as follows:

the proportion of the primary industry is higher than the national average [41-42]

See the highlighted text on page 3 (lines 49-50) for the changes in the manuscript. 

The tertiary industry in the Yellow River Basin accounted for 44.4%, lower than the national average of 53.3% and the Yangtze River Basin of 54.1%. The tertiary industry in the Yellow River Basin only accounts for 23.4% of the national tertiary industry, which is on a downward trend. The industrial development in the Yellow River Basin has yet to form its distinct advantages, and the development level of the tertiary industry needs to be further improved.

See the highlighted text on page 4 (lines 5-11) for the changes in the manuscript.

Comment 2: Why this paper adopted the particular method SE-SBM Model? There are other methods there, what is your justification for your methodology?

Response: This study uses the SE-SBM Model to estimate the high-quality development efficiency of the Yellow River Basin. According to your comments, the authors respond from the following aspects.

  1. What other methods are currently used to measure efficiency? What are their characteristics and weaknesses? The advantages and disadvantages of FSA and DEA are mentioned in the manuscript. The specific contents are as follows:

In terms of efficiency measurement methods, the Stochastic Frontier Analysis (SFA) proposed by Aigner et al. (1977) is often used as a measure of efficiency [18]. Since SFA needs to assume the specific form of the production function, most researchers prefer the data envelopment method (DEA), which introduces the input-output slack problem to avoid bias in the model setting [19]. However, the non-radial and non-angular SBM model proposed by Tone (2001) found that this method easily ignores environmental factors and cannot be comprehensively measured by comprehensive indicators [20]. Although these research methods have certain shortcomings, they also lay a research foundation for measuring high-quality development efficiency.

See the highlighted text on page 2 (lines 38-46) for the changes in the manuscript.

  1. The characteristics and advantages of the SE-SBM model to measure the efficiency of high-quality development.

The authors provide additional explanations for the reasons for choosing the SE-SBM Model. The added contents are as follows:

Studies on environmental regulation, industrial structure, and high-quality development efficiency rarely study the nonlinear relationship among them. This study adopts the SE-SBM Model that introduces slack variables, which can be solved in a larger feasible region. We consider the undesired output in the development of the river basin while also addressing the efficiency ordering problem between effective decision-making units and comprehensively and dynamically measures the high-quality development efficiency of the Yellow River Basin.

See the highlighted text on page 3 (lines 21-28) for the changes in the manuscript.

Comment 3: Measurement indicators only adopt Zhang's (2021) work, why?

Response: Thanks for your comment. When constructing the high-quality development efficiency index system, the authors referred to the Outline of the Yellow River Basin Ecological Protection and High-quality Development Plan issued by the Central Committee of the Communist Party of China and the State Council. However, in the manuscript, we mainly cite the article by Zhang et al. This is because Zhang's research team has been working on high-quality development, urban efficiency, governance dilemmas, and other fields, and the articles he publishes are authoritative and representative. In addition, the development of the Yellow River Basin mainly focuses on high-quality development, and few studies focus on the research on high-quality development efficiency. Therefore, the literature in the manuscript lists only the article by Zhang et al. However, the authors further searched for relevant studies and supplemented the literature. For the supplementary references, see [44-46].

Comment 4: The same holds true for the pannel regression model, why choose it?

Response: Thanks for your comment. The panel regression method was used in this study for the following reasons.

(1) The constructed panel regression model can analyze whether there is a significant relationship between the independent and dependent variables and the influence strength of multiple independent variables on a dependent variable. The analysis results play an essential role in further analyzing the impact of industrial structure, environmental regulation, and their interaction terms on the high-quality development efficiency of the Yellow River Basin.

(2) The panel regression model can solve the problem of missing variables. This study introduces control variables such as the level of opening to the outside world, population density indicators, and economic development level, which can more comprehensively analyze the factors affecting the high-quality development efficiency of the Yellow River Basin.

(3) The panel regression model can realize the regression analysis of the whole watershed and sub-basin of the Yellow River Basin and can more comprehensively analyze the impact of environmental regulation, industrial structure and their interaction terms and other control variables on the high-quality development efficiency of the Yellow River Basin. The complete analysis results provide policy recommendations for improving the high-quality development efficiency of the whole basin and sub-basins.

(4) The constructed panel regression model can be used for threshold regression analysis. Considering the non-economic nature of environmental regulation and the economy of industrial structure, there is a nonlinear relationship between the two in theory, which lays the foundation for the subsequent construction of the threshold model.

Based on your comment, the authors include the necessity of constructing a panel regression model for this study in this section. The added contents are as follows: 

Panel regression models are widely used to analyze whether there is a significant relationship between independent variables and dependent variables and can also analyze the strength of the influence of multiple independent variables on a dependent variable [47-48]. This study uses panel regression to explore the correlation between each variable and the high-quality development efficiency of the Yellow River Basin, laying the foundation for constructing the threshold model in the following.

See the highlighted text on page 5 (lines 2-7) for the changes in the manuscript.

Comment 5: There are format issues that need to be proofread, for instance, pp2 "A limited number of pieces of literature discuss the high-quality development efficiency of cities and economic belts [Error! Bookmark not defined.,8]."

Response: Thanks for your reminder. The references may be garbled or cannot be displayed due to different versions of word software. The authors cited and annotated all references and proofread all formatting in the manuscript.

Reviewer 2 Report

This paper proposes solutions and suggestions according to the high-quality development efficiency of the Yellow River Basin in China from the perspective of the threshold effect. It is of great significance.

However, there are still some problems in the article, and I hope that it can be revised accordingly.

1. Abstract: Add one or two recommendations in the Abstract.

2. Introduction: The international background of environmental regulation and industrial structure should be introduced.

3.No literature review in the manuscript.

4. It is noted that the manuscript needs careful editing, paying particular attention to English grammar, spelling, and sentence structure so that the goals and results of the study are clear to the reader.

5. The font size of figures and tables should be smaller than the text font and the same as the font size of words in figures and tables. It must be maintained in the whole document that the label style and font are distinct. Next, each figure in the article must ensure the appropriate description for readers. Before submitting a revision, ensure your material is properly prepared and formatted.

6. References: Most References are in Chinese; add more references in English.

Author Response

Dear Reviewer,

Thank you for your comments concerning our manuscript entitled “The impact of environmental regulation, industrial structure, and interaction on the high-quality development efficiency of the Yellow River Basin in China from the perspective of the threshold effect." Those comments are all valuable and helpful for revising and improving our paper. We have studied the comments carefully and have made corrections that we hope meet with your approval. Revised portions are marked in yellow on the paper. The primary corrections to the paper and point-to-point responses to your comments are as follows: 

Responses to the reviewer’s comments:

Reviewer #2: 

Comment 1: Abstract: Add one or two recommendations in the Abstract.

Response: Thanks for your comment. The article's Abstract mainly introduces the research background, method, and results. However, due to the word limit of the abstract, the relevant suggestions are omitted. Based on your comments, the added contents are as follows:

such as reasonably improving the level of regional imports and exports, introducing high-quality foreign capital,

See the highlighted text on page 1 (lines 26-27) for the changes in the manuscript.

Comment 2: Introduction: The international background of environmental regulation and industrial structure should be introduced.

Response: According to your suggestion, the following three points have been added to the manuscript: 1. The evolution and characteristics of international environmental regulation. 2. Analysis of the importance of environmental regulation. 3. Environmental regulation can affect industrial structure through screening and spillover effects. The added contents are as follows:

The international community has gradually realized the importance of environmental regulation. For example, the United States has controlled pollution from the source [4] and began to gradually transition from the implementation of mandatory order regulations to incentive policies in the form of rewards and punishments. Japan and Germany arouse the protection awareness of the public and enterprises through incentive mechanisms to achieve longer-term and adequate environmental protection [5-6]. China has incorporated environmental regulation into government performance assessment, including energy conservation, emission reduction, and environmental protection in the performance assessment of local leaders, which can more effectively promote energy conservation, emission reduction, and environmental protection [7].

See the highlighted text on page 2 (lines 3-13) for the changes in the manuscript.

Comment 3: No literature review in the manuscript.

Response: Thanks for your comment. We do not include the literature review as a separate part of the manuscript but review the existing studies covering research topics and methods in the Introduction. In the Introduction, the manuscript not only expounds on the research purpose of this paper and the analysis of the current situation at home and abroad but also sorts out and summarizes the relevant literature at home and abroad. Furthermore, references have been added based on other experts' opinions to enrich the Introduction's content.

   The authors supplemented the review of the existing research in the Introduction, which mainly includes three aspects: 1. Efficiency measurement often ignores the problem of undesired output. 2. In the research, the interaction between environmental regulation and industrial structure is rarely used to influence the path and mechanism of high-quality development efficiency. 3. The nonlinear relationship between the three is rarely studied. The added contents are as follows:

In the current research, there are still the following problems: the efficiency measurement often needs to pay attention to the problem of undesired output; the interaction term between environmental regulation and industrial structure is rarely used as the active path that affects the efficiency of high-quality development. Studies on environmental regulation, industrial structure, and high-quality development efficiency rarely study the nonlinear relationship among them.

See the highlighted text on page 3 (lines 18-23) for the changes in the manuscript.

Comment 4: It is noted that the manuscript needs careful editing, paying particular attention to English grammar, spelling, and sentence structure so that the goals and results of the study are clear to the reader.

Response: Thank you for your comments. We have proofread the manuscript for grammar, spelling, and sentence structure, hoping to understand this study's purpose and findings better. For example, the title of 3.3 is changed to Results and Analysis of Threshold Model Regression.

Comment 5: The font size of figures and tables should be smaller than the text font and the same as the font size of words in figures and tables. It must be maintained in the whole document that the label style and font are distinct. Next, each figure in the article must ensure the appropriate description for readers. Before submitting a revision, ensure your material is properly prepared and formatted.

Response: The authors have adjusted the font size and style of all the figures and tables in the article and added supplementary explanations to the missing parts to ensure a more specific understanding of the figures and tables. We analyzed the characteristics of the curve change in Figure 1. Based on the results in the figure, we also compare the trends in different watersheds and explore the reasons for the differences and changes. See the highlighted text on page 7 (lines 12-32) in the manuscript.

Comment 6: References: Most References are in Chinese; add more references in English.

Response: Thank you for your suggestion. References are cited in the Introduction and the discussion of the results in several places, but there are slightly more references in Chinese. The authors have added and replaced some references in the manuscript.

The revisions are as follows: in the references, new references 6-7; references 17; references 21; references 33; references 43-44; references 47-48;

references 11; references 14-15; references 23; references 28-29; references 35; references 37; references 40; references 50-51; references 53; references 55; references 58 are replaced.

See the highlighted text on page 14 (references) for the changes in the manuscript.

Round 2

Reviewer 2 Report

I am satisfied with the revised version.

The data in this manuscript are reliable, the research methods used are reasonable, and the results obtained are robust and interesting.

I think this paper is suitable for publication in IJERPH now.